# Socio-ecological factors influencing dietary behaviours among adolescents and young adults in rural Eastern Uganda: A qualitative study

Thomas Buyinza[1,2]*, Edward Buzigi[3,4], Joshua Kitimbo[3], Gabriel Ssabika[2,5], Mary Mbuliro[1], Julius Kiwanuka[1], Justine Bukenya[3], David Guwatudde[1], Rawlance Ndejjo[6,7]

1 Department of Epidemiology and Biostatistics, School of Public Health, Makerere University, Kampala, Uganda, 2 Iowa State University-Uganda Program, Center for Sustainable Rural Livelihoods, Iowa State University, Ames, Iowa, United States of America, 3 Department of Community Health and Behavioural Sciences, School of Public Health, Makerere University, Kampala, Uganda, 4 Department of Public Health & Nutrition, Faculty of Health Sciences, Victoria University, Kampala, Uganda, 5 Department of Food Technology and Nutrition, School of Food Technology, Nutrition and Bioengineering, Makerere University, 6 Department of Disease Control and Environmental Health, School of Public Health, Makerere University, Kampala, Uganda, 7 Department of Preventive Medicine, College of Medicine, Korea University, Seoul, South Korea

* thomasbuyinza@gmail.com

## Abstract

### Introduction

Adolescents and young adults (AYAs) worldwide display poor dietary behaviors, including high consumption of sugar-sweetened beverages and insufficient intake of fruits and vegetables. These issues are more pronounced in Sub-Saharan Africa, such as rural Eastern Uganda, where 45.3% of adolescents eat low-diversity diets high in refined grains and fats. Such diets raise the risk of malnutrition and diet-related non-communicable diseases (NCDs). However, there is limited contextual evidence on the multi-level factors influencing AYAs' dietary behaviors in rural Uganda. This study examined socio-ecological factors shaping dietary behaviors among AYAs in this setting.

### Methods

A qualitative study guided by the socio-ecological model (SEM) was conducted in Mayuge District, Eastern Uganda. Focus group discussions (FGDs) were held with AYAs, including male and female, aged 10–14, 15–19, and 20–24 years. To have a nuanced understanding of how AYAs' dietary behaviours are shaped, additional FGDs were conducted with parents or guardians and teachers, and key informant interviews with the district education officer, nutrition focal person, civil society staff, and food vendors. Data were analyzed in ATLAS.ti using both inductive and

**Data availability statement:** All de-identified data analyzed for this manuscript are provided as S4 File.

**Funding:** This study was conducted as part of the ARISE-NUTRINT project, a collaborative project between Makerere University School of Public Health (MakSPH) in Uganda, and 13 other institutions in Africa, Europe and the Harvard School of Public Health in the United States of America. The project is funded by the European Union under Project Grant Number 101095616. The funders had no role in the study design, data collection and analysis, decision to publish, or preparation of the manuscript. Funding acquisition at MakSPH was led by DG and JB, co-Investigators on the ARISE-NUTRINT project.

**Competing interests:** All authors declare no competing interests.

deductive thematic approaches: data-driven sub-themes were first identified inductively, then deductively mapped onto pre-determined themes of the SEM.

## Results

Dietary behaviors were shaped by satiety, energy needs, sensory appeal, and nutrition knowledge at the individual level. Peer influence, parental control, and food's perceived link to attractiveness acted interpersonally, while community factors included gendered cultural taboos, norms, and health worker advice. At the societal level, cultural identity, ancestral restrictions, and media exposure strongly influenced choices.

## Conclusions

This study contributes novel rural-specific evidence from rural Uganda, where AYAs' diets are uniquely constrained by satiety demands, parental dominance, cultural taboos, and seasonal scarcity; contrasting with urban contexts where convenience, autonomy, and wider food environments prevail. Multi-level interventions integrating nutrition education, family and peer engagement, cultural dialogue, and household food security support are essential for promoting healthier diets in resource-limited rural settings.

## Introduction

Adolescents and young adults (AYAs) aged 10–24 years [1] comprise approximately 1.9 billion people, about 24% of the global population [2]. The vast majority (90%) live in low- and middle-income countries (LMICs), including those in Sub-Saharan Africa (SSA) [2] where they constitute over 60% of the total population [3]. Adolescence is characterized by an increase in body requirements for calories and essential nutrients such as vitamins and proteins [4]. During adolescence, increased requirements for calories and essential nutrients coincide with rapid physical and cognitive maturation [4]. At the same time, social and developmental changes increase susceptibility to external influences such as peer norms and social approval [5], with AYAs often prioritizing immediate rewards over long-term health benefits [6,7]. Consequently, AYAs are particularly drawn to poor-quality diets such as refined grains, processed meat, and snacks high in fats or salt [8–10].

In 2023, the prevalence of unhealthy dietary behaviors among adolescents globally stood at 50% of adolescents consuming sugar-sweetened beverages (SSBs) daily, 37% having inadequate fruit intake, and 28.5% lacking sufficient vegetable consumption [11]. These poor dietary practices are more pronounced in SSA, particularly among adolescents of low socio-economic status [12,13]. In 2020, 34% of AYAs in South Africa consumed SSBs at least five times per week [14]. Similar trends were observed in Ethiopia and Ghana, where over 51% of in-school adolescents had diets low in fruits and vegetables and high in sugars, leading to inadequate intake of

essential nutrients [15,16]. Across Uganda in 2023, 89.6% of young people aged 18–29 years consumed fewer than five servings of fruits and vegetables daily [17]. In 2020, among 498 adolescents in Iganga and Mayuge in Eastern Uganda, 45.3% had low dietary diversity (consumed fewer than four of the nine standard food groups in the past 24 hours); 87% consumed fats or oils [18]. In the same setting (Mayuge district), 12.6% of 1,206 AYAs had overall poor diet quality in 2025 [19]; where 29.2% consumed refined grains and flour products more than four times per week, 19.8% and 32.1% consumed purchased deep-fried foods, SSBs, two to three times per week, respectively. Only 23.6% consumed leafy vegetables or fruits more than four times per week [19].

These poor-quality dietary patterns negatively impact adolescents' growth and development [20–22], a critical concern during their rapid phase of physical and cognitive maturation in adolescence. Besides, such diets predispose adolescents to complications related to micronutrient deficiency, like anemia, or under-nutrition, including wasting, underweight, or stunting [23], and overnutrition, including overweight and obesity [22–27]. Over time, poor dietary habits, overweight, or obesity can increase adolescents' lifetime risk of developing nutrition-related non-communicable diseases (NCDs), especially type-2 diabetes mellitus and cardiovascular diseases [24,28].

Despite growing recognition of the role of socio-ecological determinants of dietary behaviors, beyond knowledge and attitudes, there remains limited contextual evidence on these multi-level factors among AYAs in rural LMICs like Uganda. While studies from urban SSA and high-income countries (HICs) have identified factors such as food access [29], social networks, including peers and family [30], and dietary norms like the stigma around vegetable consumption [31], much of this evidence comes from women of reproductive age [31], adolescents in urban Uganda [32], or adolescents in North Africa [33]. However, these findings may not be generalizable to rural Ugandan settings due to differing socio-cultural, economic, and ecological realities. To address this gap, our study explored the socio-ecological factors influencing dietary behaviors among AYAs in Mayuge District, rural Eastern Uganda.

## Methods

### Study design and setting

A qualitative study was conducted as part of the ARISE-NUTRINT initiative (Africa Research, Implementation Science, and Education – Reducing nutrition-related NCDs in adolescence and youth: Interventions and policies to boost nutrition fluency and diet quality in Africa) [34,35]. The ARISE-NUTRINT study seeks to enhance understanding of dietary and physical activity-related risks for NCDs among AYAs in seven SSA countries of Uganda, Tanzania, Burkina Faso, South Africa, Ethiopia, Nigeria, and Ghana, with partners from Europe and North America [34,35].

This paper focuses exclusively on the Uganda site of the initiative, conducted within the Iganga-Mayuge Health and Demographic Surveillance Site (HDSS) [36]. This HDSS is predominantly rural, spanning Iganga, Mayuge, and Bugweri districts, with approximately 18,634 households and a population of about 120,000 residents, 27% being adolescents [36]. Mayuge District, where data for the current study were collected, depend on subsistence farming and small-scale trade as the main sources of livelihood [36] and a 2025 study in this area documented poor dietary behaviours among AYAs [19].

### Study population and sampling

The study population comprised AYAs, their parents or guardians, teachers, *basket food vendors (community members who carry baskets containing ready-to-eat foods such as fried cassava, roasted groundnuts, fruits, etc., and sell them to students around schools)*, and key district- and community-level stakeholders in Mayuge district. Purposive sampling was used to obtain diverse perspectives on AYAs' dietary behaviors. For AYAs, maximum variation was ensured by sex (male or female), age group (10–14, 15–19, 20–24 years), schooling status (in-school or out-of-school), and residence within the Iganga–Mayuge HDSS catchment informed the selection of AYAs. Out-of-school AYAs were approached directly at households, while in-school AYAs were approached at schools, with local leaders (LC1) facilitating mobilization but not influencing participation.

Parents or guardians of these AYAs, their teachers, and basket food vendors at the respective schools were also purposively recruited. Additional stakeholders included the district education officer, district nutrition focal person, and representatives of civil society organizations (CSOs) involved in adolescent health and nutrition programming. No purposively selected participants declined or dropped out.

A total of twelve FGDs and six KIIs were conducted. Eight FGDs were held with AYAs, stratified by sex (male or female), age group (10–14, 15–19, and 20–24 years), and schooling status (in-school or out-of-school) to capture diverse perspectives. Two FGDs were conducted with parents or guardians (separate for male and female), and two with teachers (primary and secondary school levels). The six KIIs involved the Mayuge District Education Officer, the District Nutrition Focal Person, two CSO staff, and two basket food vendors. The final sample size was guided by data saturation, which was reached when subsequent discussions or interviews yielded no new insights.

## Conceptual framework

This study was underpinned by the social-ecological model (SEM), which posits that health behaviors, such as diet, result from the interaction of factors across individual, interpersonal, organizational, community, and societal levels [37]. At the individual level, the study examined how gender, nutrition knowledge, skills, attitudes, and personal preferences shape dietary behavior. The interpersonal level considered the role of parents, peers, social networks, household income, and food security. Community and organizational-level factors included resources, socioeconomic status, cultural norms, and local food environment. At the societal and environmental level, broader structural, cultural, and contextual elements that shape behavior at a macro level were explored. These included environmental conditions (like food availability and seasonality), local infrastructure and access, media and marketing factors, and entrenched cultural belief systems or taboos not necessarily enforced by immediate social ties.

By applying the SEM, this study adopted a multi-layered perspective to explore the diverse and interconnected factors influencing dietary behaviours among AYAs in rural Eastern Uganda, as illustrated in the hypothesized conceptual framework (S1 File).

## Data collection

Guides for FGD and KII informed by literature review were used to collect data from participants, recruited from 17th June 2024–30th August 2024. The guides were pretested among non-participants in Bugweri district, a setting with socio-demographic characteristics similar to Mayuge District, to assess clarity, cultural appropriateness, and question flow. Feedback from the pretest informed minor revisions to wording and sequencing to enhance contextual relevance and comprehensibility. The finalized guides, translated into *Lusoga*, the primary local language; contained open-ended questions tailored to each participant group. Examples included: "What factors influence your food and drink choices?" (FGDs with AYAs); "What is your opinion about the eating habits of young people in this community?" (KII with food vendors); and "What factors shape the dietary choices of AYAs in the district?" (KII with CSO and district staff). The complete guides are provided in S2 File. Prior to each KII and FGD sessions, basic socio-demographic information (e.g., age, education level, area of residence, and school status for the AYAs) was collected from each participant.

All KIIs were conducted by three male research team members (TB, JKit, and JKiw), each fluent in English and *Lusoga*, holding Master of Public Health degrees, and with prior experience in qualitative data collection. At the time of the study, they were serving as Research Associates at the School of Public Health, Makerere University. This team was complemented by a female Research Assistant (JW), with Bachelors of Counselling and Guidance, with extensive experience in nutrition and qualitative data collection, and fluent in both *Lusoga* and English. KIIs with basket food vendors were conducted in *Lusoga* while interviews with CSO representatives and district staff were conducted in English. All interviews were conducted face-to-face in private and convenient locations; primarily participants' workplaces or community venues,

to ensure confidentiality and limit access by non-participants. Each KII lasted 45–60 minutes. To strengthen interpretation of societal-level factors influencing dietary behaviours, three repeat KIIs (two with district staff and one with a CSO representative) were conducted to clarify selected sub-themes.

All FGDs were conducted face-to-face in Lusoga, moderated by two male members of the research team (TB and JKit) alongside the female Research Assistant (JW). Each FGD involved an average of eight participants and was held in private, neutral venues such as schools or health centres, identified with support from local community health workers and local council one chairpersons. One researcher moderated the discussion, while another managed audio recording and took field notes. Both FGDs and KIIs were audio-recorded using digital voice recorders, with detailed field notes taken to capture non-verbal cues and contextual observations. Each FGD lasted 60–90 minutes.

Although the researchers were not members of the study communities, they were familiar with rural Eastern Ugandan contexts and AYAs' nutrition issues, which facilitated meaningful engagement with participants. To maintain neutrality, no prior personal relationships existed between researchers and participants. Recruitment occurred in multiple settings, with guidance from community leaders: out-of-school AYAs and parents were approached through household visits; in-school AYAs, basket food vendors, and teachers during school visits; and district staff and CSO officials at their workplace offices. At the start of each interaction, researchers introduced themselves as public health professionals conducting a study on factors influencing dietary behaviors among AYAs, with the aim of informing future nutrition improvement programs. The study objectives, confidentiality procedures, and voluntary nature of participation were clearly explained prior to obtaining consent.

Regular debriefing meetings were held between the data collectors and another member of the research team (JB), a senior faculty at the School of Public Health, Makerere University, to review field experiences. The senior member provided close supervision and ongoing support, offering feedback on interview techniques and guidance on addressing emerging challenges. These efforts helped ensure consistency and enhance the overall quality of data collection.

## Data management and analysis

Audio recordings from the FGDs and KIIs were transcribed verbatim and simultaneously translated into English by three research team members (TB, JKit, and GS). To ensure accuracy and fidelity to participants' responses, another team member (MM) reviewed each transcript while cross-checking with the original audio recordings. All personal identifiers were removed during transcription to maintain confidentiality. To gain a deep understanding of the data, four of the team members (TB, JKit, GS, and JKiw) read the transcripts at least three times and independently documented initial impressions. These insights informed the development of a codebook, which was collaboratively discussed, refined, and harmonized to ensure consistency during coding.

Transcripts were imported into ATLAS.ti version 9, and coding was conducted using a combination of inductive and deductive approaches. Inductively, the team identified key concepts, ideas, and patterns expressed directly by participants, particularly those related to factors influencing AYA dietary behaviours. Recurring codes were organized into sub-themes, guided by the phases of thematic analysis as outlined by Braun and Clarke [38]. The research team reviewed the sub-themes to ensure coherence and relevance, merging, splitting, or excluding them as appropriate. Final codes and sub-themes were summarized in a data master sheet and deductively mapped onto the pre-determined themes: the levels of the SEM (individual, interpersonal, community, organizational, and societal).

The analysis is presented as a narrative synthesis, supported by typical quotations by participants, codes, and sub-themes, to describe the socio-ecological factors influencing dietary behaviours among the AYAs. The study methods and findings are reported in accordance with the COnsolidated criteria for REporting Qualitative Research (COREQ) guidelines [39], and the completed COREQ checklist is provided in S3 File.

## Ethical consideration

Ethical clearance for ARISE-NUTRINT project was obtained from the Research and Ethics Committee of the School of Public Health, Makerere University (Ref: SPH-2023–460), and the study was registered with Uganda National Council for Science and Technology (Ref: HS3481ES). Written informed consent was obtained from parents or guardians of adolescents aged 10–17 years, followed by assent from the younger adolescents themselves. Participants aged 18 years and above provided their own written informed consent. Collected data contained no personally identifiable information and was stored securely on password-protected devices and servers, with access limited to authorized research personnel only.

## Results

The total number of participants was 102, including 64 AYAs, 16 parents or guardians, 16 teachers, two food vendors, two civil society representatives, and two district officials (education officer and nutrition focal person). Among the AYAs, 32 were out of school, four had attained tertiary education, and 27 were from households with 5–10 members. All 16 parents or guardians were aged 25 years and above; six had at least secondary education, 11 were married or cohabiting, and only two were formally employed. All 16 teachers had attained tertiary education, and 13 of them were married or cohabiting. Detailed participant characteristics are presented in Table 1.

### Socio-ecological factors influencing dietary behaviours among AYAs

This section describes the findings of the study in terms of sub-themes, supported by participants' quotes. We present the sub-themes, organized based on the pre-determined themes of the SEM, as shown in the empirical framework (S5 File).

**Individual level factors.** At the individual level, the ability of food to satisfy hunger or provide energy, sensory affection, and nutrition knowledge were key factors influencing dietary behaviour among AYAs in rural Eastern Uganda.

### Satisfying hunger and providing energy

Adolescents and young adults commonly associated certain foods with the ability to satisfy hunger and provide energy, reflecting a strong belief in the functional benefits of eating. However, rather than referencing specific nutrients or nutritional knowledge, participants described these choices in terms of the extent to which food made them feel strong, satisfied, and fueled them to complete daily tasks.

Both male and female AYAs identified foods like posho (stiff maize meal) and beans as "energy-giving," particularly useful for physically demanding work such as digging, riding boda-bodas (motorcycle taxi) or household chores:

> *"What motivates us as boys is the food that gives us strength and energy in our bodies. For me, when I eat like posho and beans, I can get energy to dig very well."* (FGD, boys, 16 years, in school)

> *"For me as a girl, what motivates me to eat a particular type of food is the desire to eat what helps my body to have energy to do my daily work and to also grow very well."* (FGD, girls, 18 years, out of school)

> *"Now that I ride a boda-boda, I eat foods that fill me fast so I can continue without feeling hungry."* (FGD, boys, 23 years, out of school)

Gendered contrasts were also observed: girls often described restrictions linked to taboos, while boys still framed food around strength and masculinity:

> *"As girls we are instructed not to eat emamba [lungfish]; elders say it can cause misfortune or even affect our ability to have children. Because of this, I don't eat it, even though I know it gives energy."* (FGD, girls, 17 years, in school)

**Table 1. Socio-demographic characteristics of participants (n = 102).**

| Characteristic | AYAs | Parents or Guardians | Teachers | Food vendors, CSOs, and District staff) | Overall |
|---|---|---|---|---|---|
| **Age group** | | | | | |
| 10-14years | 21 | 00 | 00 | 00 | 21 |
| 15-19years | 22 | 00 | 00 | 00 | 22 |
| 20-24years | 21 | 00 | 00 | 00 | 21 |
| 25 and years and above | 00 | 16 | 16 | 6 | 38 |
| **School status** | | | | | |
| In-school | 32 | 00 | 00 | 00 | 32 |
| Out of school | 32 | 16 | 16 | 6 | 70 |
| **Highest education attained** | | | | | |
| Primary | 34 | 8 | 00 | 02 | 44 |
| Secondary | 26 | 06 | 00 | 00 | 32 |
| Tertiary | 04 | 02 | 16 | 04 | 26 |
| **Household size** | | | | | |
| < 5 people | 26 | 05 | 10 | 02 | 43 |
| 5-10 people | 27 | 09 | 06 | 04 | 46 |
| >10 people | 11 | 02 | 00 | 00 | 13 |
| **Marital status** | | | | | |
| Single never married | 47 | 00 | 00 | 00 | 47 |
| Married/staying with partner | 15 | 11 | 13 | 4 | 43 |
| Separated/Divorced | 02 | 05 | 03 | 02 | 12 |
| **Disability** | | | | | |
| Yes | 07 | 00 | 00 | 00 | 07 |
| No | 57 | 16 | 16 | 06 | 95 |
| **Employment** | | | | | |
| Formal | 00 | 02 | 16 | 04 | 22 |
| Informal | 23 | 14 | 00 | 02 | 39 |
| Not working | 41 | 0 | 00 | 00 | 41 |

## Sensory affection: taste, aroma, and appearance

Taste, aroma, and visual appeal acted as powerful, interrelated motivators of food choice among AYAs, influencing not only what they ate but also how they emotionally and socially related to food. Across group discussions, AYAs consistently highlighted the importance of taste, particularly a preference for sweet, oily, and strongly flavored foods that are readily accessible within their communities. For many, taste was a primary criterion in deciding what to eat:

> *"I always first consider the kind of food that is tasty and sweet but not harmful."* (FGD, boys, 12 years, in school).

> *"Some of those healthy foods smell badly for me, like greens (i.e., amaranth leaves, cabbage..). That's why I fail to eat them."* (FGD, girls, 14 years, in-school)

Adolescents and young adults were drawn to street foods and quick snacks that delivered both sensory satisfaction and ease of access. Parents corroborated these views, frequently observing their children's inclination toward street foods such as *rolex* (a popular snack made of fried eggs rolled with chapati), plain chapati, sodas, and energy drinks, foods known for their rich flavor profiles and availability:

*"Most young people prefer eating rolex, plain chapati, sodas, and energy drinks."* (FGD with parents, female, 38 years)

Teachers also noted that adolescents' food preferences are highly influenced by preparation methods that enhance taste. Foods perceived as bland or less flavorful, such as unfried vegetables, were often rejected in favor of tastier options:

*"Whenever we cook unfried green vegetables, they [AYAs] refuse to eat food because they want it fried. They say that unfried food is not tasty."* (FGD with teachers, secondary school)

Beyond taste, aroma played a similarly influential role in food decisions. For many AYAs, the smell of food acted as a trigger for cravings, prompting impulsive food-seeking behaviors. The scent of familiar street foods, particularly chapatis, was frequently cited as an irresistible sensory cue:

*"For instance, you can be walking by the roadside and you smell the nice chapati aroma. You are then forced to look for the money to buy it."* (FGD, boys, 13 years, in school)

Visual appeal further deepened this sensory connection. In particular, adolescent girls emphasized the importance of food presentation and appearance in stimulating their interest and willingness to eat. For some, visual appeal could override even concerns about how the food was prepared:

*"What motivates me is the food that appeals to my eyes, whether well prepared or not."* (FGD, girls, 18 years, out of school)

**Nutrition knowledge**

Nutrition knowledge was an important factor shaping AYAs' dietary behaviours, with stronger knowledge observed among in-school adolescents compared to their out-of-school peers.

Among in-school AYAs, school-based nutrition education was frequently cited as the main source of information about healthy eating practices:

*"The teacher at school taught us that we have to eat a balanced diet so that we are healthy, so for me, I always try to eat a balanced diet."* (FGD, boys, 17 years, in school)

By contrast, those who were out-of-school demonstrated limited awareness of balanced diets, underscoring disparities linked to educational access:

*"I just eat what is there… mostly cassava and dry tea [black tea]. I don't know much about which foods are healthy."* (FGD, boys, 20 years, out of school)

Key informants reinforced these subgroup patterns, emphasizing that low nutrition knowledge remains a barrier to healthy eating especially among out-of-school AYAs in Mayuge district:

*"So, low level of knowledge about a balanced diet and nutritious foods to eat negatively affects the food choices of most of the out-of-school AYAs here in Mayuge district."* (KII, CSO staff)

**Interpersonal level.**  At the interpersonal level, perceived role of food in enhancing physical and sexual appeal and eating to belong (peer and social pressures), parental decision, and keeping up with the neighbours' plate, were the key factors found to influence dietary behaviours.

## Perceived role of food in enhancing physical and sexual appeal

Across discussions with AYAs, teachers, and district stakeholders, distinct subgroup patterns were identified, describing how AYAs perceived the role of food in shaping physical attractiveness and sexual appeal. Girls, especially younger (10–14 years) and in-school – framed food choices around body image and sexual appeal, while boys, particularly out-of-school youth, linked food to energy and work productivity.

Among younger adolescent girls aged 10–14 years, eating sweet foods was described as a way to enhance body image and appear attractive to peers:

> *"I also want my body to look nice and sexually appealing, so I have to eat sweet things, which helps me gain weight and improve skin colour to look good."* (FGD, girls, 13 years, in school)

Older adolescent girls aged 15–19 years who were out of school expressed similar views, linking fatty foods with beauty and social acceptance:

> *"For us girls, when you eat foods that make you grow fat, people admire you and say you are beautiful, so I also try to eat such foods when I can."* (FGD, girls, 19 years, out of school)

By contrast, young boys aged 20–24 years who were out of school linked food choices with productivity and strength needed for physical work, to earn income:

> *"For us boys, when we eat posho and beans, we get the energy for do heavy work like making bricks or loading sugar canes. It is about having energy to keep working to make money."* (FGD, boys, 20 years, out of school)

Teachers confirmed that these beliefs were widespread among in-school adolescents, particularly older adolescents (15–19 years), who frequently equated sweet or "good" foods with attractiveness and desirability:

> *"They [AYAs] prefer eating good and sweet things. Like at my school, they say that if you eat sweet things, you also become sexually appealing… that is their reason, and that is how they understand it. During that adolescent stage, when they eat sweet things, they believe they will also become sexually appealing to their partners."* (FGD with teachers, primary school)

Key informants emphasized that these pressures were gendered and disproportionately influenced girls. A district health official highlighted a locally held myth:

> *"For example, there is a myth among girls here in Mayuge District that if they eat a lot of fatty foods and grow fat, they will be loved by boys because of their enhanced beauty."* (KII, district health staff, Mayuge)

## Eating to belong: Peer and social pressures

Across interviews and group discussions, participants consistently highlighted how AYAs' dietary behaviours were shaped by a web of peer and social pressures. These pressures reflected a quest for social belonging, identity, and status. Both boys and girls framed food in relation to acceptance within friendship groups. However, boys also make certain food choices to gain status or favour in peer and romantic relationships (reported by food vendors).

In-school adolescents described conforming to the eating patterns of their peers as a way of blending into social groups and avoiding exclusion:

*"The motivation comes from the rest of the people they are with because they are also eating, so that they can fit in the group."* (FGD, boys, 18 years, in school)

Adolescent girls emphasized the pressure of pocket money and the social expectation to consume certain foods or drinks alongside friends:

*"When my friends have pocket money, they buy fried cassava and pancakes. If I don't also buy, they laugh at me."* (FGD, girls, 13 years, in school)

District education officials reinforced this point, noting that peer influence was particularly pronounced in school settings, where adolescents adopted their friends' preferences for trendy or "fast" foods:

*"The food choices of some adolescents to eat fast food stuff like chapatis, chips, etc. influence the food choices of their peers as well."* (KII, district education staff, Mayuge)

A basked food vendor cited that adolescents, particularly those with access to money, sometimes use food as a tool to gain social favor or impress peers and lovers. The food vendors also highlighted that AYAs prioritize eating out, often buying more expensive or locally trendy foods to signal status or affection.

*"Whenever these young people get some money, they prefer eating out of home. Some of them have started getting girlfriends who ask for chicken, and the boy has to excite her by buying the roasted chicken."* (KII, basket food vendor at secondary school).

### Parents decide, AYAs eat

Across adolescents, parents, and key informants (CSOs), there was a shared recognition that food choices within households were largely dictated by parents or caregivers, with AYAs having limited input. The AYAs described a lack of autonomy, often consuming whatever food was available or prepared, especially by mothers:

*"Usually, we eat what our parents prepare, what is available at home. Sometimes during the sweet potato harvest season, you find that we eat sweet potatoes because it's what our mother always prepares."* (FGD. boys, 14 years, in school)

This dynamic was reinforced by key informants, who pointed to structural and cultural norms that assign decision-making around food to adults rather than children:

*"In most cases, children and adolescents have limited power to select the food they would like to eat. They eat whatever is available in the household, as decided by parents."* (KII, CSO staff)

Parents themselves openly acknowledged this authority. They emphasized that food preparation was primarily guided by seasonal availability and household economic constraints, and that children were expected to accept what they [parents] affords to provide:

*"As parents, we usually prepare food based on what we have harvested or can afford to buy; and I always tell my children to learn to eat what is available."* (FGD with parents, male, 41 years)

**Keeping up with the neighbours' plate**

Neighbours' choices were frequently cited as shaping AYAs' dietary preferences. Adolescents described how observation and emulation influenced their food aspirations, often leading them to desire meals similar to those consumed by peers and surrounding households:

> *"I can look at the neighbors and then desire food they have eaten, then we also go back home and decide to eat what they have eaten."* (FGD, boys, 18 years, in school).

This form of social mirroring extended beyond adolescents themselves, creating pressure within households. Parents explained that their children's requests were often directly linked to what they saw others preparing in the neighbourhood:

> *"Whenever my children see my neighbors cooking vegetables, they come back and ask me to buy for them too."* (FGD with parents, female, 49 years,)

For many parents, meeting such demands became a way of maintaining harmony at home and avoiding disappointment, even when these requests added to household expenses:

> *"I buy vegetables so that I don't disappoint them."* (FGD with parents, female, 49 years,)

**Community and organizational level.** Gendered cultural food taboos and norms and reception of health workers were the two sub-themes identified as factors influencing food choices, under this theme: -

**Gendered cultural food taboos and norms**

At the community level, gendered cultural norms influenced AYAs' dietary choices. Traditional taboos, particularly those targeting girls and women – restricted the consumption of specific foods such as *emamba* (lungfish) and chicken. These norms were widely upheld by community actors, including parents, elders and food vendors.

Adolescent girls emphasized early socialization into these restrictions, often describing them as rules they had to follow from childhood:

> *"As a girl, since I was young, my grandmother told me never to eat emamba. Even now at my age, I can't try it because people will say I disobeyed our culture."* (FGD, girls, 16 years, out-of-school)

By contrast, boys highlighted their freedom to consume *emamba* and pointed out that the restriction was gender-specific:

> *"We boys can eat emamba freely, but girls are not allowed. For us, it makes us strong, but for them it is a taboo."* (FGD, boys, 18 years, out of school)

Food vendors reinforced these cultural norms, reflecting how deeply embedded they were in community practices:

> *"From childhood, girls in this community are taught not to eat emamba (lungfish); it is considered taboo for me or any other girl or woman. As a result, girls avoid it; even if they might want to try it."* (KII, basket food vendor at a secondary school)

### Reception of health workers' advice

Adolescents and young adults positioned healthcare providers as authoritative voices in shaping their dietary practices. Participants reported that guidance from health workers strongly influenced their food choices, especially in the context of maintaining health and preventing disease. The legitimacy of this guidance was rarely questioned, with many youths acknowledging the value of medical advice in navigating food decisions:

> *"Doctors advise that we eat food that will soften our digestive system, and that's what influences me when making food choices."* (FGD, boys 12 years, in school).

Health concerns, particularly chronic or acute illnesses, were also reported as major triggers for dietary modifications. Some adolescents described avoiding certain foods based on diagnoses or health risks identified by health professionals:

> *"Then diseases or sickness, like you can want to take some sugar, but when you are sick with diabetes, that cannot allow you to eat."* (FGD, boys, 16 years, in school).

In addition to direct interaction with health professionals, adolescents frequently applied nutrition knowledge conveyed through caregivers and community health promotion efforts. This mediated form of health education appeared to enhance their perceived ability to make informed food choices, especially in contexts where formal education or access to health services was limited:

> *"Whenever I get the opportunity, I cook greens because I was told by a community health worker it helps to prevent diseases."* (FGD, girls, 14 years, in school).

**Society and environmental level factors.** At the societal and environmental level, the dietary behaviours of AYAs were shaped by broader structural and contextual factors that extended beyond their immediate social interactions. These included cultural identity and ancestral food taboos, media exposure and celebrity endorsement, food seasonality and physical access. These factors operated at a macro level, intersecting with socioeconomic and geographic contexts, and were consistently highlighted across AYAs, parents, civil society, and district-level informants.

### Cultural identity and ancestral food taboos

At the societal level, deep-rooted cultural identities and ancestral food taboos shaped dietary behaviors across ethnic and clan lines. Among AYAs, these taboos were described as inherited prohibitions tied to clan beliefs and symbolic meanings of certain foods. For some, foods considered totems were completely avoided, regardless of their nutritional value. These restrictions were perceived not as personal choices but as obligatory observances tied to cultural heritage:

> *"As a Musoga (a member of the Basoga ethnic group in eastern Uganda), I come from a clan that prohibits eating certain foods, especially those considered to be totems. We also eat a lot of sweet potatoes; it's part of our culture."* (FGD, boys, 19 years, in-school)

From the perspective of district-level stakeholders, such dietary patterns reflected distinct ethnic affiliations within the region. Specific staples were associated with different groups; *Banyole, Basoga*, and *Baganda* – highlighting how ethnic food traditions continued to shape AYAs' food preferences at a population level:

*"Among the ethnic groups in this area, the Banyole mostly eat millet bread; the Basoga prefer cassava and sweet potatoes; while the few Baganda living here tend to eat matooke [steamed green bananas]. Young people in Mayuge generally follow these culturally rooted food preferences."* (KII, district staff, Mayuge)

**Media and celebrity endorsements**

Media exposure such as watching television (TV) and celebrity endorsements were identified as modern, trans-local forces that shaped AYAs' aspirations around food and lifestyle. These influences were described as aspirational and identity-driven, often clashing with locally available foods and varied across gender, age, and schooling status.

Adolescent boys aged 15–19 years, reported being especially drawn to foods they saw in movies or advertisements, associating them with desirability and social status:

*"I love eating the kind of food I see in movies; it always looks so delicious."* (FGD, boys, 16 years, in school).

Female AYAs expressed similar sentiments, with many saying they were motivated to try new foods after seeing celebrities endorse them:

*"When I see my favorite artist eating something, I feel like I want to try it too."* (FGD, girls, 12 years, in school).

Among out-of-school young adults, media influence often reached them indirectly through peers rather than direct access. Males described relying on friends with smartphones to share what they saw online. Relatedly, females echoed similar aspirations, particularly around foods linked to beauty and celebrity status:

*"I don't watch TV much, but when my friends in town who have smartphones tell me about foods they see on Facebook or TikTok, I also feel like trying them."* (FGD, boys, 22 years, out of school).

*"Sometimes my friends talk about the foods celebrities eat, like fried chicken or pizza. Even if I haven't seen a pizza myself, but wish to taste it one day."* (FGD, girls, 20 years, out of school).

These pressures were also acknowledged by parents, who described difficulties in managing their children's exposure to food advertising:

*"My child often asks for foods they've seen on television. It's hard to say no when they insist."* (FGD with parents, male, 35 years).

Teachers highlighted the pervasiveness of media influences, noting that they often shaped classroom discussions or snack preferences among students:

*"Students often talk about foods they've seen on TV or their favorite celebrities eating. It definitely affects their choices."* (FGD with teachers, secondary school).

**Food seasonality based on rainfall patterns**

Food seasonality was an overarching factor shaping dietary behaviours, with AYAs, parents, and teachers alike recognizing that seasonal shifts determined what foods were grown, available, and ultimately consumed. For AYAs in rural communities, dietary behaviours fluctuated with what was locally produced during specific seasons, often limiting dietary diversity:

> "*During dry seasons, we mostly have cassava bread and sweet potatoes. Those are the easiest foods to get,* and that is what they [AYAs] eat*"* (FGD for parents, female, 45 years).

Teachers also echoed this view, indicating that even school environments are affected by seasonal dynamics, which influence what learners bring from home and what schools provide. District staff corroborated these patterns, emphasizing the constrained choices that seasonality imposes on AYAs:

> "*When it's sweet potato season, that's what they eat; when it's maize season, the same applies. Even greens depend on availability.*" (FGD, teacher, secondary school).

> "*The availability of food in a given season largely determines what adolescents eat. Their diet changes with the season.*" (KII, district staff, Mayuge).

Furthermore, institutional settings such as schools often had limited flexibility to respond to seasonal variations. School feeding programs tended to rely on staples like posho and beans; foods that are logistically practical but nutritionally monotonous:

> "*Most schools provide posho and beans; some fry them, some don't; but it's what is consistently available.*" (FGD for teachers, secondary school).

### Local availability and physical access

Adolescents and young adults' food choices were strongly influenced by their geographic location and the physical proximity of food outlets. Participants emphasized that regional agricultural production shaped what foods were both culturally consumed and practically accessible. For instance, adolescents highlighted how staple foods varied across areas:

> "*People in Mbarara have bananas because they grow a lot of them. In Busoga, it's mostly sweet potatoes; it depends on where you come from.*" (FGD, boys, 15 years, in school).

Physical distance from food outlets further restricted access, particularly for in-school boys, who reported challenges in obtaining purchased foods such as beverages due to mobility constraints:

> "*Even when you have money, it's hard to get juice if the shop is too far from where you are.*" (FGD, boys, 18 years, in school).

District officials reinforced these perspectives, noting that AYAs largely consumed what was grown locally within their immediate surroundings:

> "*Adolescents in Mayuge mostly eat what is grown locally. They rarely have the option to choose from a wide variety of foods.*" (KII, district staff, Mayuge).

### Discussion

This study explored socio-ecological factors influencing dietary behaviors among AYAs in rural Mayuge, Eastern Uganda. Unlike much of the existing evidence, which is predominantly drawn from urban SSA settings [29,40,41] and HICs [42,43], our study offers rural-specific insights where income levels, food systems and environments differ substantially. For

example, AYAs in rural settings often contend with seasonal food availability and subsistence farming [18,36], limited dietary options [15], and stronger parental or cultural controls over food decisions [44]. In contrast, urban adolescents typically face broader exposure to convenience foods [41], diverse retail outlets [40], and greater individual autonomy in food choices [45]. Positioning our findings within the rural context allows us to identify both unique multi-levels drivers of AYAs' dietary choices, such as satiety needs amplified by seasonal scarcity; and shared sociocultural influences, like gendered food taboos, peer and media pressures, thereby bridging a critical evidence gap which can inform design of context-appropriate interventions.

At the individual level, satiety and energy provision were consistently emphasized as primary drivers of food choices. AYAs described seeking additional meals from nearby trading centers to remain full. Unlike AYAs in urban Uganda, where convenience and time-saving often dominate food decisions [41], AYAs in the rural context of our study were more concerned with the physical functionality of food: ability of the food to fill them and providing body energy for labor-intensive daily activities. Seasonality further amplified these patterns: during dry seasons when diets were monotonous and less diverse, satiety-driven choices became even more central, illustrating how environmental constraints intersect with physiological needs. Beyond energy, sensory appeal: taste, aroma, and appearance; was another potent factor. AYAs in our study expressed preferences for oily, fried, and flavorsome foods, paralleling findings from urban Benin [46], and Ghana [47]. Yet in rural Eastern Uganda, where diets are dominated by staples and seasonal crops [18,19], the craving for sensory-rich foods may partly explain low fruit and vegetable intake, highlighting the need for strategies that adapt traditional diets into more appealing yet nutritious forms [48].

Nutrition knowledge also shaped dietary behaviors among AYAs in rural Uganda, with marked differences by school status. In-school AYAs frequently cited teachers as their primary source of information, while their out-of-school counterparts demonstrated limited understanding of balanced diets, which likely contributed to the reported poorer food choices. These patterns mirror evidence across urban SSA, where low food knowledge among young women has been linked to unhealthy eating [41], and where formal education and socioeconomic status influenced adolescent dietary behaviors, as observed in Ghana [49] and Kenya [50]. Further evidence from SSA reinforces the importance of nutrition knowledge in shaping food practices [31,51]. Collectively, these findings underscore the need to integrate practical, skills-based nutrition education into school curricula while also reaching out-of-school AYAs, through community-based education programs, ensuring that knowledge is effectively translated into healthy and sustainable dietary behaviors.

At the interpersonal level, both body image perceptions and peer influence strongly shaped dietary practices. Teachers and health officials noted that girls were thought to consume sweet and fatty foods to enhance physical and sexual attractiveness, while boys often equated larger body size with strength and masculinity. These findings echo perceptions from Eastern Uganda, where being overweight was associated with beauty, prosperity, and peace of mind [52], and align with global evidence linking food to sexual appeal and desirability [53]. Media exposure reinforced these ideals: celebrity endorsements and viral digital content promoted processed and energy-dense foods, consistent with Western studies connecting digital marketing to unhealthy diets [54,55]. In the same vein, cultural and media pressures combined with peer influence acted as powerful forces, in our study, with AYAs reporting the consumption of meat, fried snacks, and sugary drinks to gain social acceptance or avoid ridicule. Similar patterns have been reported in urban SSA [47,56]. Access to pocket money further intensified these peer dynamics, illustrating a cross-level interaction between household economic conditions and social expectations that reinforced unhealthy dietary behaviors.

Parental dominance was another critical determinant in rural Mayuge, Uganda. Mothers decided which food to prepare for the household, based on seasonal availability and income, leaving AYAs with limited autonomy. This differs from urban LMICs [29,40] and HICs [42,43] contexts where AYAs wield greater decision-making power due to financial independence and media-driven exposure. Gender norms further reinforced parental control: for instance, girls were disproportionately restricted by food taboos (such as prohibitions against eating lung fish), mirroring findings from Tanzania [44]. These

intersections between gender norms, parental authority, and cultural taboos highlight the layered constraints shaping diets among AYAs in rural Uganda.

Community and organizational influences included the trust placed in both facility-based and community health workers. AYAs reported modifying diets – such as reducing sugar or eating soft foods; based on provider advice, reflecting strong reliance on professional credibility, which mirrors findings in South Africa [48,57]. In contexts of limited facility access, caregivers and community health workers mediated this advice, as similarly observed in Nigeria [58] and Ghana [59]. Building on these community-level dynamics, broader societal factors also shaped dietary behaviors. Food seasonality acted as a structural influence, with AYAs in rural Eastern Uganda reporting monotonous diets during dry seasons dominated by cassava bread or sweet potatoes, a pattern also documented in Ethiopia [15]. Importantly, these environmental constraints intersected with satiety-driven preferences, amplifying the prioritization of filling foods over dietary diversity. Cultural identity and gendered food taboos further restricted choices, while media exposure reinforced existing ideals and unhealthy preferences. Together, these community and societal influences highlight the need for multi-level interventions that leverage trusted health providers while simultaneously addressing seasonal food insecurity, cultural norms, and media-driven pressures.

Overall, this study demonstrates that dietary behaviors of AYAs in rural Uganda are shaped by multi-level and intersecting factors, including satiety needs amplified by seasonal scarcity, peer norms strengthened by economic access, gender norms reinforced by parental control and taboos, and media influences amplifying body image ideals. These rural-specific dynamics contrast with urban SSA and HICs contexts, where convenience, autonomy, and wider food environments dominate. By systematically unpacking these cross-level interactions, our findings not only highlight the novelty and rural specificity of Mayuge in Eastern Uganda but also contribute critical insights for designing interventions that are contextually grounded, culturally sensitive, and responsive to the lived realities of AYAs in rural SSA.

### Study limitations and strengths

A primary limitation of this study was the reliance on self-reported data, which may have been influenced by participants' perceptions or social desirability, potentially leading to misreporting of factors influencing dietary behaviors. However, this limitation was mitigated through methodological triangulation. The study engaged a diverse group of participants, including AYAs, parents, food vendors, teachers, the district nutrition focal person, and representatives from civil society organizations involved in adolescent health and nutrition. Additionally, responses were cross-validated through a combination of FGDs and KIIs. Probing techniques were used during data collection to enhance clarity, encourage honesty, and reduce social desirability bias, thereby improving the credibility and reliability of the findings.

### Conclusion

This study shows that dietary behaviors among AYAs in rural Mayuge, Eastern Uganda are shaped by interconnected influences across individual, interpersonal, cultural, and broader societal levels. Unlike urban SSA or HICs contexts, where convenience and autonomy often dominate, rural adolescents' food choices were strongly driven by satiety and energy needs, parental control, gendered cultural taboos, peer and media pressures, and seasonal scarcity. By applying the socio-ecological model, our study provides novel rural-specific evidence from SSA, addressing a critical gap in adolescent nutrition research.

Improving AYAs' diets in such contexts requires multi-level interventions: strengthening nutrition education within schools; promoting peer- and health worker – led outreach that involves parents; supporting household agro-initiatives to diversify production and counter myths tied to body image; and engaging cultural leaders to challenge restrictive norms. At community and societal levels, empowering health workers and improving food system resilience to seasonal scarcity are

also essential. Future research should examine how these dietary determinants evolve over time and evaluate scalable, context-sensitive interventions to sustainably improve adolescent diets and reduce malnutrition and NCD risks in rural Uganda and similar settings.

## Supporting information

**S1 File. Conceptual framework.** Conceptual framework illustrating the socio-ecological determinants of dietary behaviors among adolescents and young adults in rural Mayuge, Eastern Uganda.
(DOCX)

**S2 File. Data collection guides.** Interview and focus group discussion guides used with adolescents, parents, food vendors, teachers, CSO staff, and district officials.
(DOCX)

**S3 File. COREQ checklist.** Completed COnsolidated criteria for REporting Qualitative research (COREQ) checklist for this study.
(DOCX)

**S4 File. Empirical framework.** Empirical framework illustrating socio-ecological factors influencing dietary behaviours among AYAs in Mayuge, rural Eastern Uganda.
(DOCX)

**S5 File. FGD and KII transcripts.** De-identified transcripts from focus group discussions and key informant interviews analyzed for this manuscript.
(DOCX)

## Acknowledgments

The authors would like to thank the adolescents and young adults, parents or guardians, teachers, food vendors, district staff and civil society members who participated in the study.

## Author contributions

**Conceptualization:** Thomas Buyinza, Edward Buzigi, David Guwatudde, Rawlance Ndejjo.

**Formal analysis:** Thomas Buyinza, Edward Buzigi, Joshua Kitimbo, Gabriel Ssabika, Julius Kiwanuka, Rawlance Ndejjo.

**Funding acquisition:** Justine Bukenya, David Guwatudde.

**Investigation:** Thomas Buyinza, Edward Buzigi, Justine Bukenya, David Guwatudde.

**Methodology:** Thomas Buyinza, Edward Buzigi, Joshua Kitimbo, Gabriel Ssabika, Mary Mbuliro, Julius Kiwanuka, Justine Bukenya, David Guwatudde, Rawlance Ndejjo.

**Project administration:** Thomas Buyinza, Mary Mbuliro, David Guwatudde.

**Resources:** David Guwatudde.

**Supervision:** Thomas Buyinza, Edward Buzigi, Justine Bukenya, David Guwatudde, Rawlance Ndejjo.

**Writing – original draft:** Thomas Buyinza.

**Writing – review & editing:** Thomas Buyinza, Edward Buzigi, Joshua Kitimbo, Gabriel Ssabika, Mary Mbuliro, Julius Kiwanuka, Justine Bukenya, David Guwatudde, Rawlance Ndejjo.

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
