## [Decision Letter · Decision Letter 0]

11 Aug 2025

Dear Dr. Buyinza,

Thank you for submitting your manuscript to PLOS ONE. After careful consideration, we feel that it has merit but does not fully meet PLOS ONE’s publication criteria as it currently stands. Therefore, we invite you to submit a revised version of the manuscript that addresses the points raised during the review process.

We look forward to receiving your revised manuscript.

Kind regards,

Okikiolu Badejo, M.D., Ph.D

Academic Editor

PLOS ONE

Journal Requirements:

2. In the online submission form, you indicated that “Due to confidentiality concerns and the potential for deductive disclosure in this qualitative dataset, the data are not publicly available. However, relevant portions of the de-identified excerpts may be shared by the corresponding author (TB) upon reasonable request and in line with the ethics approval conditions.”

4. We note that “Supplementary File 2.” in your submission contain copyrighted images. All PLOS content is published under the Creative Commons Attribution License (CC BY 4.0), which means that the manuscript, images, and Supporting Information files will be freely available online, and any third party is permitted to access, download, copy, distribute, and use these materials in any way, even commercially, with proper attribution. For more information, see our copyright guidelines: http://journals.plos.org/plosone/s/licenses-and-copyright.

1. You may seek permission from the original copyright holder of “Supplementary File 2.” to publish the content specifically under the CC BY 4.0 license.

Reviewers' comments:

Reviewer's Responses to Questions

**Comments to the Author**

1. Is the manuscript technically sound, and do the data support the conclusions?

Reviewer #1: Yes

Reviewer #2: Yes

2. Has the statistical analysis been performed appropriately and rigorously?

Reviewer #1: No

Reviewer #2: N/A

3. Have the authors made all data underlying the findings in their manuscript fully available?

Reviewer #1: Yes

Reviewer #2: Yes

4. Is the manuscript presented in an intelligible fashion and written in standard English?

Reviewer #1: Yes

Reviewer #2: Yes

Reviewer #1: The qualitative results in this manuscript would help contextualize dietary choices among AYA in a rural region within an LMIC, making this research informative.The manuscript is well structured with a detailed and well contextualized introduction setting a strong basis for the scope and expected outcomes for the rest of the manuscript.However the authors will need to improve the methods section by providing details on their sampling approach. Qualitative studies are not necessarily expected to be generalisable so subjects selection must be systematic. Authors could add the justification for convenience sampling if that was the approach they chose to use. Further review details are below.

Major comments:

Line 93: Clarify Population based qualitative study, does this refer to your sampling frame being derived from the DHSS or the generalizability of your data

Line 94: Other than age and sex what informed your sampling frame for AYA and how does this frame reflect the earlier assertion of a population based qualitative study. Were participants selected from various regions in Mayuge. Were there other criteria informing sampling , how did the community leaders decide who to reach out to and recommend them for recruitment into the study or was this convenience sampling. Could you explain it was not possible to use a more robust sampling technique?

Line 113:Provide details on participant selection among the purposively selected participants. For example, were parents/guardians related to the AYA in the FGDS or were these just parents in the community, what informed their selection?

Minor comments

Line 72: “Only 23.6% consumed leafy vegetables/fruits 4+ times/week” ; Use sentence structure and replace symbols with words for proper flow. Do this across the document replacing symbols with words (except in the abstract) and removing unnecessary capitalization.

Line 120: Basket foot vendors is likely a contextually specific name, define it please or use a more common term.

Line 549-558: Is not supported by your results as the two people who spoke about this were KII and not AYA . It would be good to point out that this perception came from the adults unless you have an AYA quote backing up this opinion.

Line 579-581 “In contrast conducted in urban LMICs or HICs often have more decision-making power at the household level, influenced by media exposure, peer trends, and access to pocket money, such as in Zambia [26] and across Africa [49].” This statement has missing information, you might want to name the population you refer to for clarity.

Line 604: the word Uganda is missing

Line 73: Overall you have strong references but Sawyer ( Ref 18) should not be the primary reference in that introduction section as that publication is a review, replacing with a primary study would be more informative.

Reviewer #2: Line 53-55: The logical progression in this section is unclear. The statement that “changes that come along with the transition from adolescence to adulthood make individuals more susceptible to emotional and social factors” requires further clarification and supporting evidence. It is not explained which specific changes are being referred to, or the mechanism by which these changes increase susceptibility. Furthermore, the link between such susceptibility and a tendency to favour short-term over long-term benefits is asserted but not sufficiently justified. Please consider elaborating on these points and providing references to support the proposed causal relationships.

Line 68: It would be helpful to briefly clarify what is meant by “dietary diversity” for readers less familiar with the term.

Line 388: Quote is missing.

General comments:

Highlight the novelty and rural specificity: Make it clearer which findings are novel or distinctive to Mayuge, and how they extend or differ from what is already known from urban SSA or HIC contexts. This will help bridge the current evidence gap on rural settings.

Clarify expected differences: When comparing rural findings to urban SSA or HIC studies, briefly explain where differences are expected, for example, due to variations in income, infrastructure, or food environments.

Make subgroup patterns more explicit: Subgroup differences are hinted at but not systematically presented. It would strengthen the discussion to explicitly outline patterns by gender, in-school vs. out-of-school status, and adolescents vs. young adults.

Elaborate more on cross-level interactions: Some important intersections are mentioned but could be more fully developed, for example:

- Seasonality amplifying satiety-driven choices.

- Peer norms combining with pocket money to shape purchases.

- Gender norms interacting with parental control and food taboos.

- Media influences reinforcing existing body ideals.

Linking these intersections explicitly across socio-ecological levels would make the multi-level analysis more cohesive and impactful.

**Do you want your identity to be public for this peer review?** For information about this choice, including consent withdrawal, please see our Privacy Policy

Reviewer #1: No

Reviewer #2: No

---

## [Author Response · Author response to Decision Letter 1]

28 Aug 2025

Dear Editor,

Kindly received attached revised submissions.

Best regards,

Thomas Buyinza

Corresponding author

---

## [Decision Letter · Decision Letter 1]

14 Nov 2025

Socio-ecological Factors Influencing Dietary Behaviours among Adolescents and Young Adults in Rural Eastern Uganda: A Qualitative Study

PONE-D-25-36829R1

Dear Dr. Buyinza,

We’re pleased to inform you that your manuscript has been judged scientifically suitable for publication and will be formally accepted for publication once it meets all outstanding technical requirements.

Kind regards,

Okikiolu Badejo, M.D., Ph.D

Academic Editor

PLOS ONE

Additional Editor Comments (optional):

Reviewers' comments:

Reviewer's Responses to Questions

**Comments to the Author**

Reviewer #1: All comments have been addressed

Reviewer #2: All comments have been addressed

2. Is the manuscript technically sound, and do the data support the conclusions?

Reviewer #1: Yes

Reviewer #2: (No Response)

3. Has the statistical analysis been performed appropriately and rigorously?

Reviewer #1: Yes

Reviewer #2: (No Response)

4. Have the authors made all data underlying the findings in their manuscript fully available?

Reviewer #1: Yes

Reviewer #2: (No Response)

5. Is the manuscript presented in an intelligible fashion and written in standard English?

Reviewer #1: Yes

Reviewer #2: (No Response)

Reviewer #1: This is a well written and structured article with a strong justification for the study. Authors have adequately addressed all reviewer comments.The methodological approach is sound and the article is laid out clearly. My remaining comment is for the authors to define within the methods section, line 106, the definition of adolescents. Once that is addressed i believe the article would be suitable for publication.

Reviewer #2: The manuscript is recommended for acceptance, as the substantive content is strong and the prior comments/feedback have been adequately addressed. However, it should undergo careful copyediting prior to final publication to polish the text and correct minor typographical and grammatical errors. Such revisions will improve overall readability and presentation.

**Do you want your identity to be public for this peer review?** For information about this choice, including consent withdrawal, please see our Privacy Policy

Reviewer #1: No

Reviewer #2: No

---

## [Editor Report · Acceptance letter]

PONE-D-25-36829R1

PLOS ONE

Dear Dr. Buyinza,

I'm pleased to inform you that your manuscript has been deemed suitable for publication in PLOS ONE. Congratulations! Your manuscript is now being handed over to our production team.

Kind regards,

on behalf of

Dr. Okikiolu Badejo

Academic Editor

PLOS ONE